# Synthesising Realistic Calcium Traces of Neuronal Populations Using GAN

## Abstract

Calcium imaging has become a powerful and popular technique to monitor the activity of large populations of neurons in *vivo*. However, for ethical considerations and despite recent technical developments, recordings are still constrained to a limited number of trials and animals. This limits the amount of data available from individual experiments and hinders the development of analysis techniques and models for more realistic sizes of neuronal populations. The ability to artificially synthesize realistic neuronal calcium signals could greatly alleviate this problem by scaling up the number of trials. Here, we propose a Generative Adversarial Network (GAN) model to generate realistic calcium signals as seen in neuronal somata with calcium imaging. To this end, we propose CalciumGAN, a model based on the WaveGAN architecture and train it on calcium fluorescent signals with the Wasserstein distance. We test the model on artificial data with known ground-truth and show that the distribution of the generated signals closely resembles the underlying data distribution. Then, we train the model on real calcium traces recorded from the primary visual cortex of behaving mice and confirm that the deconvolved spike trains match the statistics of the recorded data. Together, these results demonstrate that our model can successfully generate realistic calcium traces, thereby providing the means to augment existing datasets of neuronal activity for enhanced data exploration and modelling.

## 1 Introduction

The ability to record accurate neuronal activities from behaving animals is essential for the study of information processing in the brain. Electrophysiological recording, which measures the rate of change in voltage by microelectrodes inserted in the cell membrane of a neuron, has high temporal resolution and is considered the most accurate method to measure spike activities (Dayan & Abbott, 2001). However, this method is not without shortcomings (Harris et al., 2016). For instance, a single microelectrode can only detect activity from few neurons in close proximity, and extensive pre-processing is required to infer single-unit activity from a multi-unit signal. Disentangling circuit computations in neuronal populations of a large scale remains a difficult task (Rey et al., 2015). On the other hand, calcium imaging monitors the calcium influx in the cell as a proxy of an action potential (Berridge et al., 2000). Contrary to electrophysiological recordings, this technique yields data with high spatial resolution and low temporal resolution (Grienberger & Konnerth, 2012), and has become a powerful imaging technique to monitor large neuronal populations. With the advancements in these recording technologies, it has become increasingly easier to obtain high-quality neuronal activity data in *vivo* from live animals. However, due to ethical considerations, the acquired datasets are often limited by the number of trials or the duration of each trial on a live animal. This poses a problem for assessing analysis techniques that take into account higher-order correlations (Brown et al., 2004; Staude et al., 2010; Stevenson & Kording, 2011; Saxena & Cunningham, 2019). Even for linear decoders, the number of trials can be more important for determining coding accuracy than the number of neurons (Stringer et al., 2019).

Generative models of neuronal activity hold the promise of alleviating the above problem by enabling the synthesis of an unlimited number of realistic samples for assessing advanced analysis methods. Popular modelling approaches such as the maximum entropy framework (Schneidman et al., 2006; Tkačik et al., 2014) and the latent variable model (Macke et al., 2009; Lyamzin et al., 2010) have shown ample success in modelling spiking activities, though many of these models re-

quire strong assumptions on the data and cannot generalize to different cortical areas. To this end, GANs have shown tremendous success in synthesizing data across a vast variety of domains and data-types (Karras et al., 2017; Gomez et al., 2018; Donahue et al., 2019), and are good candidates for modelling neuronal activities. Spike-GAN (Molano-Mazon et al., 2018) demonstrated that GANs can model neural spikes that accurately match the statistics of real recorded spiking behaviour from a small number of neurons. Moreover, the discriminator in Spike-GAN is able to learn to detect which population activity pattern is the relevant feature, and this can provide insights into how a population of neurons encodes information. Ramesh et al. (2019) trained a conditional GAN (Mirza & Osindero, 2014), conditioned on the stimulus, to generate multivariate binary spike trains. They fitted the generative model with data recorded in the V1 area of macaque visual cortex, and the GAN generated spike trains were able to capture the firing rate and pairwise correlation statistics better than the dichotomized Gaussian model (Macke et al., 2009) and a deep supervised convolution model.

Nevertheless, the aforementioned deep generative models operate on spike trains which are discrete in nature, and back-propagation on discrete data remains a difficult task (Caccia et al., 2018). For instance, Ramesh et al. (2019) used the REINFORCE gradient estimate (Williams, 1992) to train the generator in order to perform back-propagation on discrete data. Still, gradient estimation with the REINFORCE approach yields large variance, which is known to be challenging for optimization (Maddison et al., 2016; Zhang et al., 2017). In addition, generating and training on binary spike trains directly introduces uncertainty as the generator has to learn the deconvolution process as well, making it an even more difficult task.

In this work, we investigate the possibility of synthesising continuous calcium fluorescent signals using the GAN framework, as a method to scale-up or augment the amount of population activity data. In addition, modelling the calcium signals directly has several advantages (a) the generator needs to learn the deconvolution process when synthesising directly on binary spike trains, hence there is additional uncertainty, which is not present for calcium signals. (b) Calcium imaging signals have inherently more information about the neuronal activities than binary spike trains. (c) Based on calcium signals with known ground-truth, calcium deconvolution algorithms can be evaluated. Hence, We devised a workflow to synthesize and evaluate calcium imaging signals, then validate the method on artificial data with known ground-truth as well as mimicking real two-photon calcium ($Ca^{2+}$) imaging data as recorded from the primary visual cortex of a behaving mouse (Pakan et al., 2018; Henschke et al., 2020).

## 2    METHODS

### 2.1    NETWORK ARCHITECTURE

The original GAN framework, introduced in Goodfellow et al. (2014), plays a min-max game where the generator $G$ attempts to generate convincing samples from the latent space $Z$, and the discriminator $D$ learns to distinguish between generated samples and real samples $X$. In this work, we use the WGAN-GP (Gulrajani et al., 2017) formulation of the loss function without the need of incorporating any information of the neural activities into the training objective:

$$\mathcal{L}_D = \mathop{\mathbb{E}}_{z \sim Z}[D(G(z))] - \mathop{\mathbb{E}}_{x \sim X}[D(x)] + \lambda \mathop{\mathbb{E}}_{\tilde{x} \sim \tilde{X}}[(\| \nabla_{\tilde{x}} D(\tilde{x}) \|_2 - 1)^2] \qquad (1)$$

where $\lambda$ denotes the gradient penalty coefficient, $\tilde{x} = \epsilon x + (1 - \epsilon)\hat{x}$ are samples taken between the real and generated data distribution.

For learning calcium signal generation, we adapted the WaveGAN architecture (Donahue et al., 2019), which has shown promising results in audio signal generation. In the generator, we used 1-dimensional transposed convolution layers to up-sample the input noise. We added Layer Normalization (Ioffe & Szegedy, 2015) in between each convolution and activation layer, in order to stabilize training as well as to make the operation compatible with the WGAN-GP framework. To improve the model learning performance and stability, the calcium signals were scaled to the range between 0 and 1 by normalizing with the maximum value of the calcium signal in the data. Correspondingly, we chose sigmoid activation in the output layer of the generator and then re-scaled the signals to their original range before inferring their spike trains.

The architecture of the discriminator in our model is largely a mirror of the generator, with the exception of the removal of Layer Normalization and instead of up-sampling the input with transposed convolution, we used a simple convolution layer. Samples generated using transposed convolution often exhibit the "checkerboard" artifacts described by Odena et al. (2016), where the output exhibits repeated patterns (usually very subtle to the eye) due to a filter being applied unevenly to the receptive field. In the context of signal generation, the discrimination could exploit the periodic artifacts pattern and learn a naive policy to reject generated samples. Donahue et al. (2019) proposed the Phase Shuffle mechanism in the discriminator to address the aforementioned issue. The Phase Shuffle layer randomly shifts the activated units after each convolution layer within $[-n, n]$, in order to distort the periodic pattern. Hence, the resulting samples constitute a more challenging task for the discriminator. Figure A.4 shows a simple illustration of the Phase Shuffle operation. In our network, we incorporated the Phase Shuffle operation, as well as using a kernel size that is divisible by the stride size, as suggested in Odena et al. (2016). We apply the Phase Shuffle operation after each convolution layer, which has led to a noticeable improvement in the generated samples. Table A.1 shows the exact architecture of our model.

## 2.2 MODEL PIPELINE

We devised a consistent model analysis pipeline to evaluate the quality of samples generated by the model, as well as its ability to generalize, in the context of neuronal population spiking activities. The complete model analysis pipeline is shown in Figure A.2.

As calcium imaging is largely being used as a proxy to monitor spiking activities, we have decided to evaluate and present the inferred spike trains instead of raw calcium signals. We used the Online Active Set method to Infer Spikes (OASIS) AR1 deconvolution algorithm (Friedrich et al., 2017) to infer spiking activities from calcium fluorescent signals. We apply OASIS to both the training data and generated data to ensure the potential bias in the deconvolution process applies to the two sets of data. We then trained both the generator and discriminator with the WGAN-GP framework (Gulrajani et al., 2017), with 5 discriminator update steps for each generator update step. We used the Adam optimizer (Kingma & Ba, 2014) to optimize both networks, with a learning rate of $\lambda = 10^{-4}$, $\beta_1 = 0.9$ and $\beta_2 = 0.9999$. To speed up the training process, we incorporated Mixed Precision training (Micikevicius et al., 2017) in our codebase. The exact hyper-parameters being used in this work can be found in Table A.2.

After inferring the spike trains from the generated calcium signals, we then measure the spike train statistics and similarities using the Electrophysiology Analysis Toolkit (Denker et al., 2018). Following some of the previous works in spike generation (Macke et al., 2009; Molano-Mazon et al., 2018; Ramesh et al., 2019), we evaluate the performance of our model with the following statistics and similarities: (a) mean firing rate for evaluating single neuron statistics; (b) pairwise Pearson correlation coefficient for evaluating pairwise statistics; (c) pairwise van-Rossum distance (Rossum, 2001) for evaluating general spike train similarity. Importantly, we evaluate these quantities across the whole population for each neuron or neuron pair and each short time interval (100 ms) and compare the resulting distributions over these quantities obtained from training data as well as generated data. We therefore validate the whole spatiotemporal first- and second-order statistics as well as general spike train similarities.

## 2.3 DATA

### 2.3.1 DICHOTOMIZED GAUSSIAN ARTIFICIAL DATA

In order to verify that CalciumGAN is able to learn the underlying distribution and statistics of the training data, we generated our own ground-truth dataset with pre-defined mean and covariance using the dichotomized Gaussian (DG) model (Macke et al., 2009). The model uses a multivariate normal distribution to generate latent continuous random variables which are then thresholded to generate binary variables representing spike trains. The DG model has mean vector and covariance matrix as free parameters. To generate data from this model, we used the sample means and sample covariances obtained from real recorded data (see Section 2.3.2). In alignment with the recorded data, we generated correlated spike trains for $N = 102$ neurons with a duration of 899 seconds and at $24\,\mathrm{Hz}$, hence a matrix with shape $(21576, 102)$. In order to obtain calcium-like signals $c$ from spike trains $s$ with length $T$, we convolved the generated spike trains with a calcium response kernel

and added noise, as described in Friedrich et al. (2017):

$$s_t = g s_{t-1} + s_t \qquad 1 \leq t \leq T \qquad (2)$$
$$c = b + s + \sigma u \qquad u \sim \mathcal{N}(0, 1) \qquad (3)$$

where $g$ denotes a finite impulse response filter, $b$ is the baseline value of the signal and $\sigma$ is the noise standard deviation. In our work, we set $g = 0.95$, $\sigma = 0.3$ and $b = 0$. We scale the signal range to the unit interval. The data is then segmented using a sliding window along the time dimension with a stride of 2 and a window size of $T = 2048$ (around 85 seconds in experiment time). We apply the segmentation procedure to both the signal and spike data, hence resulting in two matrices with shape $(9754, 2048, 102)$. Examples of signals and spikes generated from the DG model can be found in Figure A.1a.

### 2.3.2 TWO-PHOTON CALCIUM IMAGING RECORDED DATA

Next, we used two-photon calcium imaging data recorded in the primary visual cortex of behaving mice. The data were collected with the same setup as specified in Pakan et al. (2018) and Henschke et al. (2020). Head-fixed mice were placed on a cylindrical treadmill, and navigated a virtual corridor rendered on two monitors that covered the majority of their visual field. A lick spout was placed in front of the mice, where a water drop would be made available to the mice as a reward if it licked at the correct location within the virtual environment. Hence, the mice would learn to utilize both the visual information and the self-motion feedback in order to maximize the rewards. Neuronal activity was monitored from the same primary visual cortex populations over multiple consecutive behavioural sessions. The basic characteristics of the recorded data are shown in Table A.3. We first experiment with calcium imaging data recorded on the $4^{\text{th}}$ day of the experiment, where the mice were familiar with the virtual environment and the given task. In this particular recording, neurons were labelled with GCaMP6f, and $N = 102$ neurons were recorded at a sampling rate of $24\,\text{Hz}$, and the mouse performed 204 trials in 898.2 seconds (raw data shape $(21556, 102)$). Due to the fact that GAN models require a significant amount of training data, information about the trial and position of the mouse in the virtual environment were not used in this work.

## 3 RESULTS

We propose CalciumGAN as a generative model to synthesize realistic calcium traces as imaged from neuronal populations. To validate our model, we used artificial data with known ground-truth as well as real data recorded from the primary visual cortex of behaving mice. We used the WGAN-GP training objective (Section 2.1) to train both the generator and discrminator. We have also experimented with the objective function from the original GAN (Goodfellow et al., 2014) and LSGAN (Mao et al., 2017). From our experiments, the WGAN-GP formulation had the best training performance and stability.

### 3.1 SYNTHETIC DATA MIMICKING DICHOTOMIZED GAUSSIAN DATA

We first fit our model with the artificial dataset sampled from the DG distribution. We trained the model for 400 epochs with 8,754 samples and held out 1,000 samples for evaluation. Since we defined the model from which we generated the training dataset, we can validate the statistics of the dataset generated by CalciumGAN on the known ground-truth directly. Examples of generated signals and its inferred spikes can be found in Figure A.1b.

Here, we compare both the trend and variation of the generated data statistics with the DG data. We estimated the mean firing rates and the covariances of data generated by CalciumGAN and compared it to the DG ones (Figure 1). We plotted the values of 5 samples for each neuron and neuron-pair, and sorted them by their mean in ascending order. The variation of the firing rate across samples matched with those of the ground-truth data. The majority of the neuron pairs have low correlation, a characteristic which was also found in the generated data. The neuron pairs that have highly positive and highly negative covariance also have a greater variation across samples.

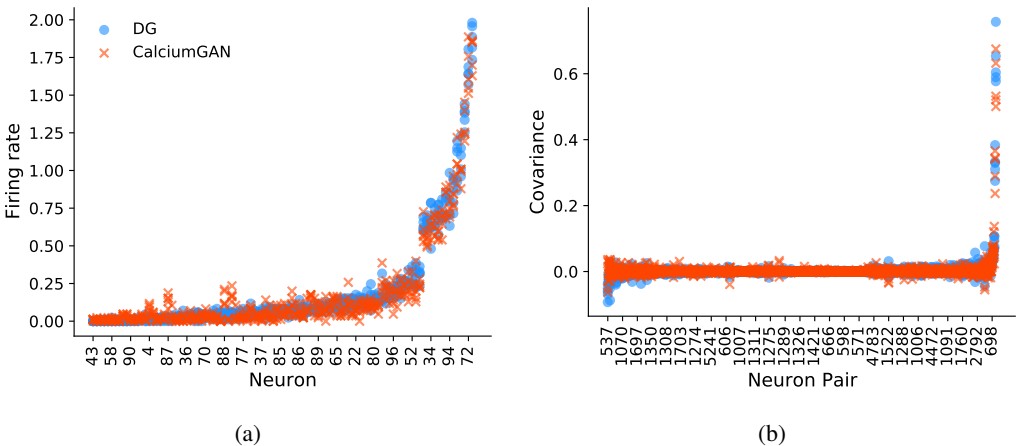

(a)                                                     (b)

Figure 1: CalciumGAN trained on the dichotomized Gaussian dataset with known ground-truth. (a) Mean firing rate of each neuron. (b) Neuron pairwise covariance. Blue dots represent DG data and orange crosses present generated data. 5 randomly selected samples for each neuron and neuron-pair were displayed in both graphs, where the order on the x-axis was sorted by the mean of the firing rate and covariance respectively. In (b), only every $10^{th}$ pair is displayed for clarity.

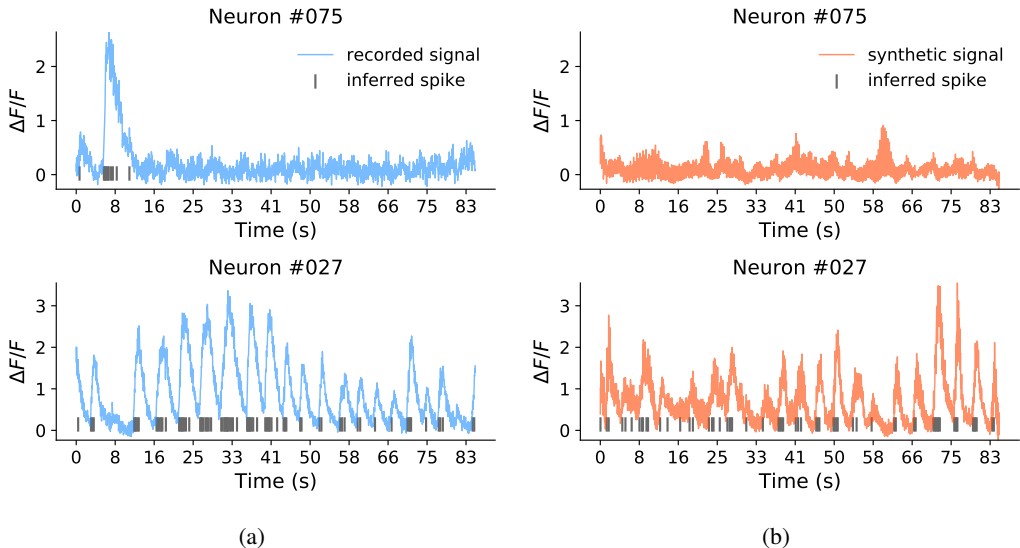

(a)                                                     (b)

Figure 2: Calcium signals and inferred spike trains (in gray) of randomly selected neurons. (a) shows the recorded data (in blue) and (b) shows synthetic data (in orange) generated by CalciumGAN trained on recorded data. **Note**: the generated data should not be identical with the recorded data, because CalciumGAN should not replicate the signals.

## 3.2 SYNTHETIC DATA MIMICKING RECORDED DATA

After validating our model on data with known ground-truth, we applied CalciumGAN on two-photon calcium imaging data recorded in the primary visual cortex of mice performing a virtual reality task. We applied the OASIS deconvolution algorithm to infer the spike activities from the recorded calcium signals, and performed the same normalization and segmentation steps as mentioned in Section 2.3.1. Figure 2a shows examples of the recorded calcium signals and inferred spike trains. There are multiple challenges for both the generator and discriminator to learn from the calcium imaging signals. Since data were segmented with a sliding window and the information of the trial was not used, some samples might consist of abnormal signal activity, such as a peak

being cropped off. Generated signals could have the same number of peaks or ranges, though might not preserve the peak and decay characteristics of calcium imaging data. Real and synthetic activity from less active neurons might be more difficult for the discriminator to distinguish due to the absence of prominent spiking characteristics.

Similar to the DG analysis, we trained the model for 400 epochs, with 8,754 training samples, and 1,000 samples were held out for evaluation. Note that since we are not taking the trial and position of the mice in the virtual environment into consideration when training the model, the generated data and the evaluation data do not have a one-to-one mapping.

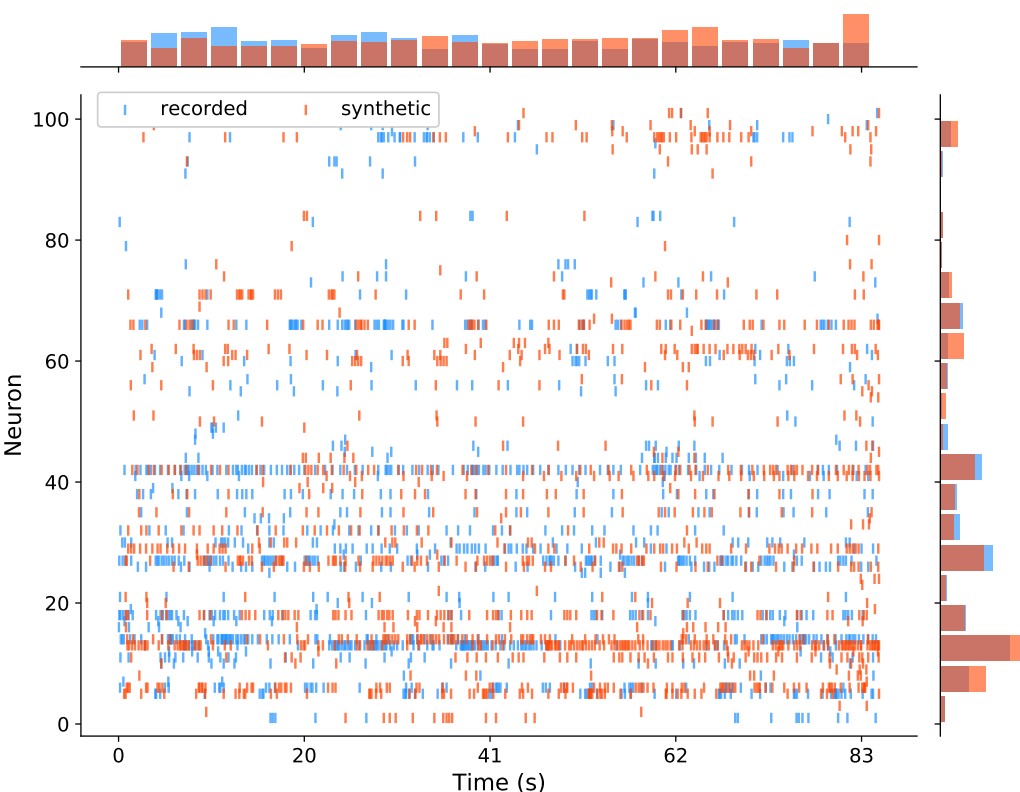

Figure 3: Raster plot of inferred real and synthetic spike trains of a randomly selected sample generated by CalciumGAN trained on recorded data. Blue markers indicate recorded data and orange markers indicate generated data. The histograms on the $x$ and $y$ axis indicate number of spikes over the temporal dimension and neuron population respectively.

We first inspect the generated data and the deconvolved spike trains visually. The calcium signals and inferred spike trains of randomly selected neurons from a randomly selected sample are shown in Figure 2b. Both the synthetic raw traces as well as the inferred spikes visually match the characteristics of the recorded ones.

We then compared the spiking characteristics across the whole population. Figure 3 shows the inferred spike trains of the complete 102 neurons population from a randomly selected sample of the real and the synthetic data, with the distribution histogram plotted on the $x$ and $y$ axis. The synthetic data mimics the firing patterns across neurons and across time remarkably well with occasional small deviations in the rates at particular temporal intervals. Notably, the samples are clearly not identical meaning that the network did not just replicate the training set data.

In order to examine if CalciumGAN is able to capture the first and second order statistics of the recorded data, we measured the mean firing rate, pairwise correlation, and van-Rossum distance (see Figure 4). The randomly selected neurons shown in Figure 4a have very distinct firing rate distributions, and CalciumGAN is able to model all of them relatively well, with KL divergence of

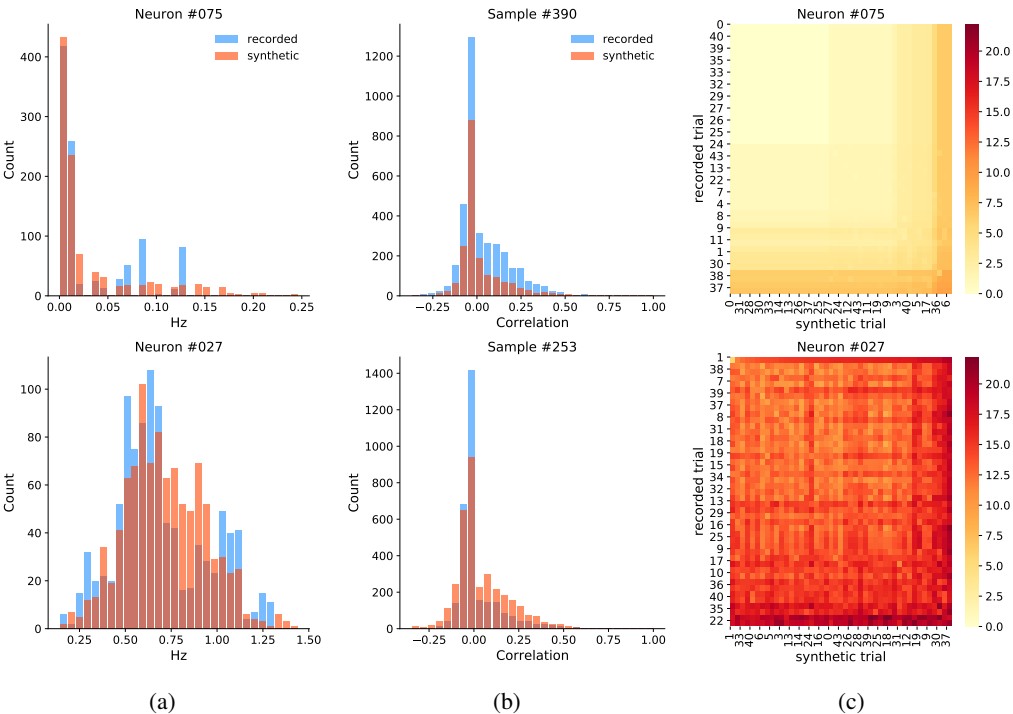

Figure 4: First and second order statistics of data generated from CalciumGAN trained on the recorded data. Shown neurons and samples were randomly selected. (a) Mean firing rate distribution over 1000 samples per neuron. (b) Pearson correlation coefficient distribution. (c) van-Rossum distance between recorded and generated spike trains over 45 samples. Heatmaps were sorted where the pair with the smallest distance value was placed at the top left corner, followed by the pair with the second smallest distance at the second row second column, and so on.

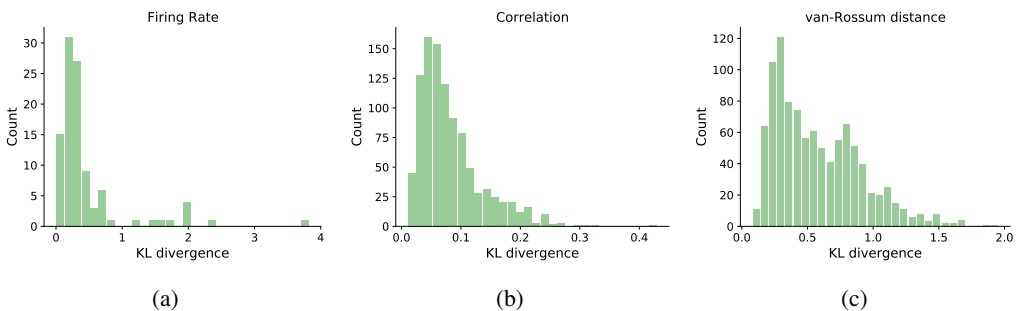

Figure 5: KL divergence of the (a) mean firing rate, (b) pairwise correlation and (c) pairwise van-Rossum distance between 1,000 recorded and generated samples.

0.31 and 0.16 with respect to the recorded firing rate over 1000 samples. We show the pairwise van-Rossum distance of the same neuron between recorded and generated data across 45 samples in Figure 4c as sorted heatmaps. Less active neurons, such as neuron 75, have a low distance value across samples, mainly due to the scarcity of firing events. Conversely, a high frequency neuron, such as neuron 27, exhibits a clear trend of lower distance values in the diagonal of the heatmap, implying the existence of a pair of recorded and generated sample that are similar. In order to ensure that the data generated by our model capture the underlying distribution of the training data, we also compute the KL divergence between the distributions of the above-mentioned metrics (see Figure 5). Note that we measure the pairwise distance of the same neuron across 50 samples in Figure 4c, whereas in Figure 5c, we measure pairwise van-Rossum distance of each neuron with respect to other neurons within the same sample. We also fitted the DG model to the recorded data

and measure the same statistics on the DG generated spike trains as a baseline. Table 1 shows the mean KL divergence of the generated data from CalciumgGAN, CalciumGAN with Phase Shuffle disabled (see Appendix A.2) and the DG model.

| Model | mean firing rate | pairwise correlation | van-Rossum distance |
|---|---|---|---|
| CalciumGAN | **0.4533** | **0.0821** | **0.5757** |
| – without Phase Shuffle | 1.0170 | 0.1027 | 0.7787 |
| DG | 1.0592 | 0.3379 | 1.0287 |

Table 1: The mean KL divergence value in each metrics of different models.

The results we presented above were trained on recordings collected from a mouse that was already familiar with the specific task. However, we were also interested in our model's capability to learn from neuronal activities that are more stochastic and potentially less correlated. To this end, we trained CalciumGAN on data recorded on the first day of the experiment (average firing rate of $58.07\,\mathrm{Hz}$ on day 1 versus $35.83\,\mathrm{Hz}$ on day 4, see Table A.3). Appendix A.3 shows the generated samples and the statistics of the inferred spike trains. The generated data were able to reflect the first and second-order statistics of the recorded data, with mean KL divergence of 0.32, 0.06 and 0.51 when comparing with the mean firing rate, pairwise correlation and van-Rossum distance, respectively. Overall, CalciumGAN was able to capture the statistics and underlying distribution of the real calcium imaging data acquired in the primary visual cortex of awake, behaving mice.

## 4 DISCUSSION

Despite the recent advancement and popularity of calcium imaging of neuronal activity in *vivo*, the number of trials and the duration of imaging sessions in animal experiments is limited due to ethical and practical considerations. This work provides a readily applicable tool to fit a GAN on calcium signals, enabling the generation of more data that matches the statistics of the provided data.

We demonstrated that the GAN framework is capable of synthesizing realistic calcium fluorescent signals similar to those imaged in the somata of neuronal populations of behaving animals. To achieve this, we adapted the WaveGAN (Donahue et al., 2019) architecture with the Wasserstein distance training objective. We generated artificial neuronal activities using a dichotomized Gaussian model, showing that CalciumGAN is able to learn the underlying distribution of the data. We then fitted our model to imaging data from the primary visual cortex of a behaving mouse. Importantly, we showed that the statistics of the synthetic spike trains match the statistics of the recorded data, without the need of incorporating any information of the neuronal activities into the model or the objective function.

We would like to highlight one potential bias in this work. To infer spike trains from the real and synthetic calcium traces, we used the OASIS deconvolution algorithm by Friedrich et al. (2017), a method which has great real-time deconvolution performance, as well as an existing Python implementation of the algorithm by the authors (Friedrich, 2017). Speed was a crucial characteristic for evaluating a large number of trials. Nonetheless, we found that this advantage often came at the cost of performance in the form of clearly missed spikes (c.f. Figure 2). However, we stress that these shortcomings apply to both the real data and the synthetic data in exactly the same way. In the end, we use the inferred spikes as a way to validate the plausibility of the synthesized traces. The comparison is fair as long as real and synthetic deconvolutions are subject to the same biases.

As the work in deep generative models continue to develop and expand, there is a limitless number of possibilities to explore at the intersection of the GAN framework and neural coding. One potential future direction for this work is to provide a meaningful interpretation for the latent generator representation. In many image generation tasks with GANs (Bojanowski et al., 2017; Karras et al., 2017) it has been shown that the output image can be modified or targeted by interpolating the latent variable that is fed to the generator. Similarly, one could potentially have final control of the generated calcium signals by exploring the synthetic calcium signals generated after interpolating samples in the latent space. Thereby, one could generate calcium imaging data that resemble the neuronal activities of an animal performing a particular novel task. Another interesting research direction would be using a GAN to learn the relationship between different neuronal populations,

or to reveal changes in activity of the same neuronal population in different training phases of an animal learning a behavioral task. This could be achieved by using, for instance, CycleGAN (Zhu et al., 2017), an unsupervised learning model that can learn the mapping between two distributions without paired data, as a potential model architecture.

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

## A    APPENDIX

| Layer | Output shape |
|---|---|
| Input | (bs, 32) |
| Dense | (bs, 2048) |
| LeakyRelu | (bs, 2048) |
| Reshape | (bs, 64, 32) |
| Conv1DTransposed | (bs, 128, 320) |
| LayerNorm | (bs, 128, 320) |
| LeakyRelu | (bs, 128, 320) |
| Conv1DTransposed | (bs, 256, 256) |
| LayerNorm | (bs, 256, 256) |
| LeakyRelu | (bs, 256, 256) |
| Conv1DTransposed | (bs, 512, 192) |
| LayerNorm | (bs, 512, 192) |
| LeakyRelu | (bs, 512, 192) |
| Conv1DTransposed | (bs, 1024, 128) |
| LayerNorm | (bs, 1024, 128) |
| LeakyRelu | (bs, 1024, 128) |
| Conv1DTransposed | (bs, 2048, 102) |
| LayerNorm | (bs, 2048, 102) |
| LeakyRelu | (bs, 2048, 102) |
| Dense | (bs, 2048, 102) |
| Sigmoid | (bs, 2048, 102) |

(a) Generator architecture

| Layer | Output shape |
|---|---|
| Input | (bs, 2048, 102) |
| Conv1D | (bs, 1024, 64) |
| LeakyRelu | (bs, 1024, 64) |
| PhaseShuffle | (bs, 1024, 64) |
| Conv1D | (bs, 512, 128) |
| LeakyRelu | (bs, 512, 128) |
| PhaseShuffle | (bs, 512, 128) |
| Conv1D | (bs, 256, 192) |
| LeakyRelu | (bs, 256, 192) |
| PhaseShuffle | (bs, 256, 192) |
| Conv1D | (bs, 128, 256) |
| LeakyRelu | (bs, 128, 256) |
| PhaseShuffle | (bs, 128, 256) |
| Conv1D | (bs, 64, 320) |
| LeakyRelu | (bs, 64, 320) |
| Flatten | (bs, 20480) |
| Dense | (bs, 1) |

(b) Discriminator architecture

Table A.1: The generator (a) and discriminator (b) architecture of CalciumGAN. The generator consists of 4,375,740 parameters, and the discriminator consists of 4,110,273 parameters. Note $bs$ denotes batch size.

| Hyper-parameters | Value |
|---|---|
| Filters | 64 |
| Kernel size | 24 |
| Stride | 2 |
| Noise dimension | 32 |
| Critic updates | 5 |
| Gradient penalty ($\lambda$) | 10 |
| Batch size (bs) | 128 |
| Epochs | 400 |
| Learning rate | 0.0001 |
| Phase shuffle (m) | 10 |

Table A.2: Hyperparamters of CalciumGAN.

| Date | Duration | Num. trials | Avg. trial duration | Avg. firing rate |
|---|---|---|---|---|
| Day 1 | 894.73 s | 129 | 6.94 s | 58.07 Hz |
| Day 4 | 898.45 s | 203 | 4.43 s | 35.83 Hz |

Table A.3: Information about the neuron population of $N = 102$ neurons recorded at $24\,\mathrm{Hz}$ from the primary visual cortex of a behaving mouse on the 1st day and 4th day of the experiment.

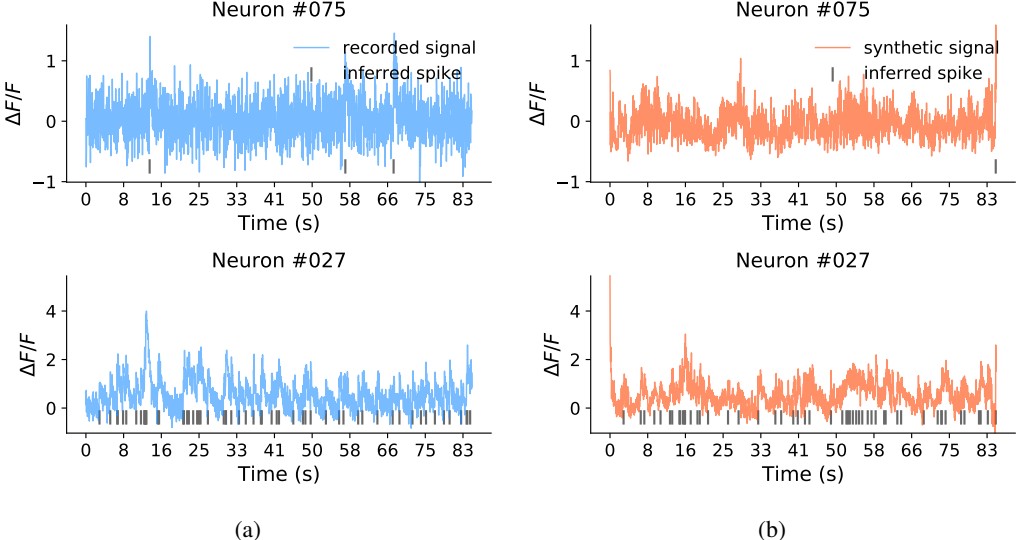

Figure A.1: Calcium signals and inferred spike trains (in gray) of randomly selected neurons. (a) shows the DG data (in blue) and (b) shows synthetic data (in orange) generated by CalciumGAN trained on the DG data. Notice that the artificial signal data transformed from DG spike data do not have the peak and decay characteristics of typical calcium imaging data. **Note**: the generated data do not incorporate the trial information, hence the generated traces do not correspond to the recorded signal in the plotted example.

## A.1 CALCIUMGAN PIPELINE

In order to train and evaluate our GAN model, we have to first pre-process the calcium signals so that they have a standardized format. For calcium imaging data of $N$ neurons with a recorded length of $L$, we would receive a raw data shape of $(L, N)$. We then used a slicing window of size $T$ to segment the data along the time dimension into $M$ segments (see Figure A.3), resulting in a matrix with shape $(M, T, N)$. To improve the network training performance, we scale the raw calcium signals $x$ to the range $[0, 1]$ before we train our generative model:

$$x_{[0,1]} = \frac{x - x_{\min}}{x_{\max} - x_{\min}} \quad (4)$$

We use $a_{[0,1]}$ to denote datum $a$ that has a range of $[0, 1]$.

After the above pre-processing step, we train CalciumGAN in mini-batches and store 1,000 samples for evaluation. Since we evaluate our model performance in terms of spike activities, we needed a deconvolution algorithm to infer the spike trains from calcium signals. In this work, we used the OASIS deconvolution algorithm (Friedrich et al., 2017) for its fast online deconvolution performance. Prior to inferring the spiking activities from the generated signals $\hat{x}_{[0,1]}$, we first have to scale the signal back to the same range as the raw calcium signals:

$$\hat{x} = \hat{x}_{[0,1]}(x_{\max} - x_{\min}) + x_{\min} \quad (5)$$

We inferred the spike trains from the generated signals as well as the real recorded data with OASIS in order to ensure the possible biases of the deconvolution algorithm are the same for both data.

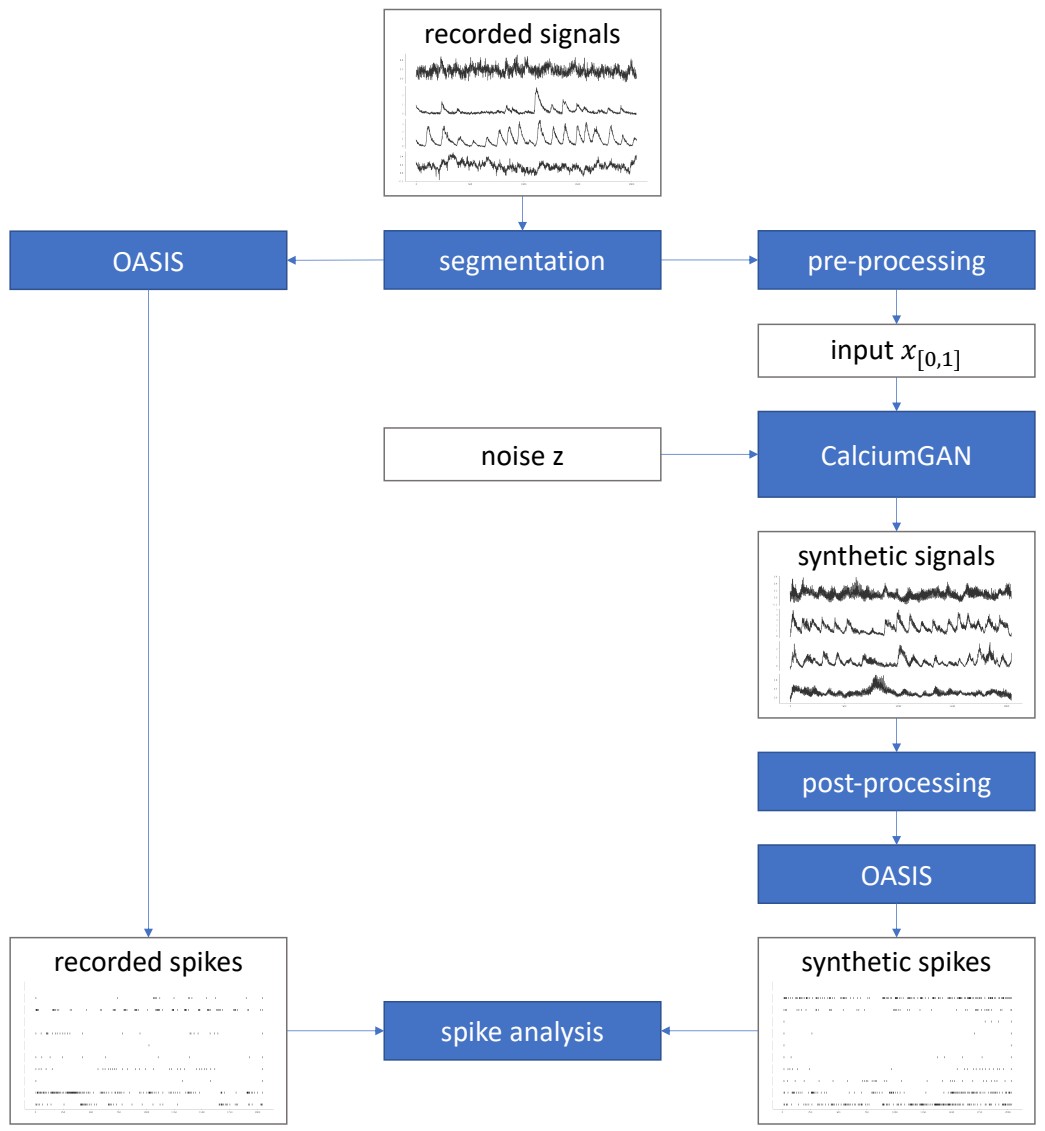

Figure A.2: Pipeline diagram of a CalciumGAN analysis. White boxes illustrate data in different processing stages. Blue boxes illustrate analysis steps and techniques.

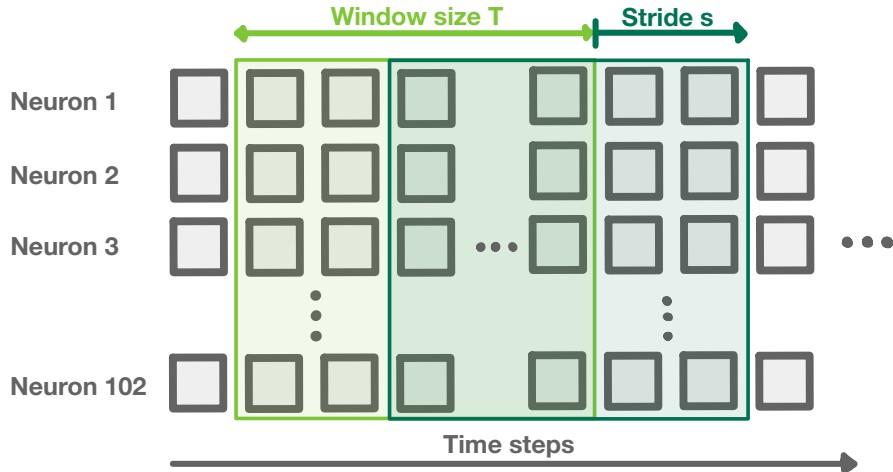

Figure A.3: Illustration of the sliding window process. The light green and green boxes represent the window with sequence length $T$ along the temporal dimension of the calcium signals. We create our training and validation dataset using a window size of $T = 2048$ and stride size of $s = 2$.

## A.2 PHASE SHUFFLE

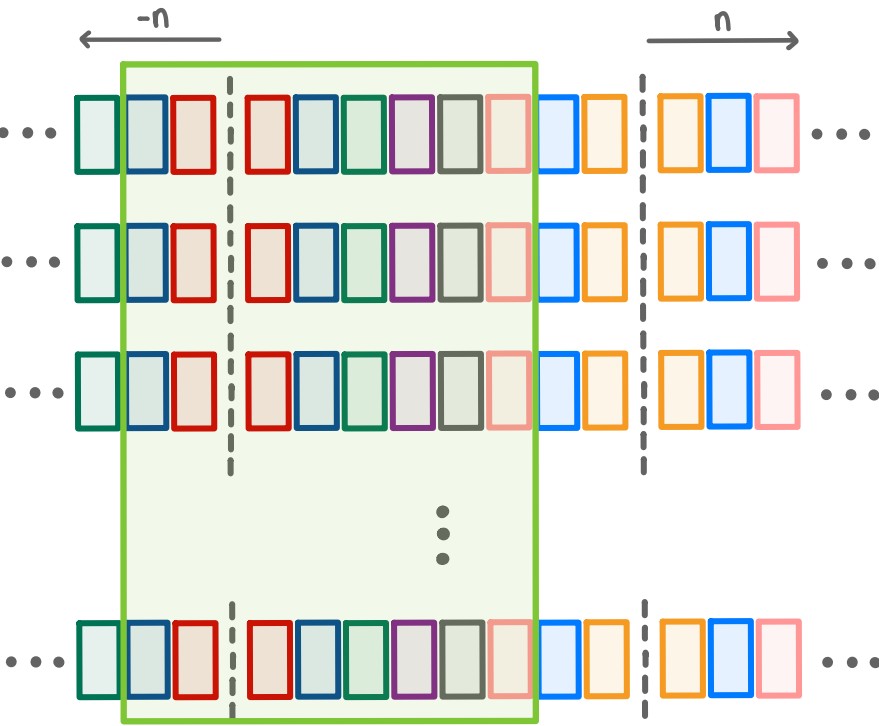

Figure A.4: Illustration of the 1-dimensional Phase Shuffle mechanism. Each box denotes an activated unit after a convolution layer, and the units are mirrored along the grey dash lines. The large box in light green represents the output of the Phase Shuffle layer with $n = -2$.

In order to reduce the effect of the "checkerboard" artifact, we adapted the Phase Shuffle mechanism (see 2.1) in the discriminator. In this section we examine the effectiveness of Phase Shuffle in terms of the visual quality of the generated traces as well as the effect it had on the inferred spike trains. A common characteristic of the calcium indicators when an action potential occur is a sharp

onset followed by a slow decay in the signal (Fröhlich, 2016). In Figure A.5, we can see that such characteristic in the calcium traces were more prominent when Phase Shuffle was enabled. We believe that such differences in the generation quality exist mainly because of the repetitive patterns in the transposed convolution layer Odena et al. (2016), since the discriminator can simply distinguish generated samples from real samples by learning if such patterns exists. As the Phase Shuffle mechanism shifts the temporal dimension (by 10 units in our experiment) randomly, it forces the discriminator to learn from other features in the data instead of the "shortcut" provided by the (undesired) nature of transposed convolution.

Moreover, not only did Phase Shuffle affect the visual quality of the generated samples, it also impacted the spike train statistics. The traces generated without Phase Shuffle lack the spiking characteristics, which made it more difficult for the deconvolution algorithm to register a spike in the data, thus increasing the inaccuracy of the inferred spike trains. When comparing the KL divergence of the spike train statistics, the samples generated without Phase Shuffle suffer worse results across the 3 statistics (see Table 1), especially with mean firing rate.

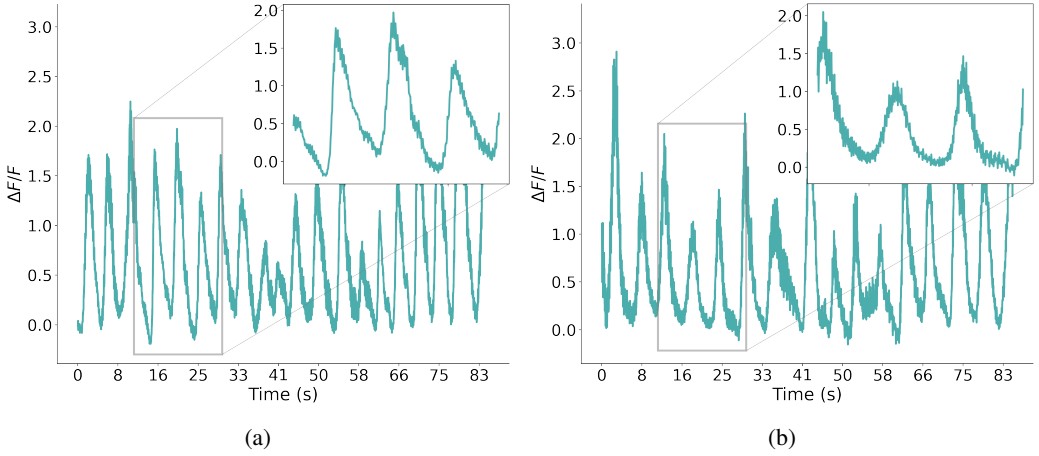

Figure A.5: Generated traces of Neuron 6 from a randomly selected sample with (a) PhaseShuffle = 10 and (b) PhaseShuffle = 0. The sharp rise to peak followed by a tail of decaying signal is less observable in when Phase Shuffle is disabled.

## A.3 DAY 1 RECORDINGS

The following figures are the generated data and spike train statistics of CalciumGAN trained on the calcium imaging recordings collected on the first day of the mice experiment.

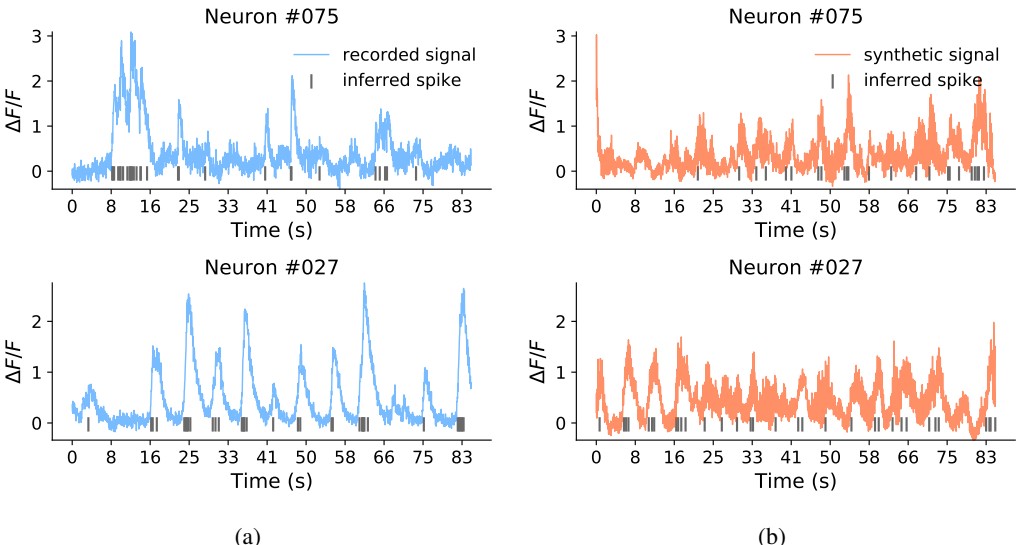

(a)                                            (b)

Figure A.6: Calcium signals and inferred spike trains (in gray) of randomly selected neurons. (a) shows the recorded data (in blue) and (b) shows synthetic data (in orange) generated by Calcium-GAN trained on recorded data. **Note**: the generated data do not incorporate the trial information, hence the generated traces do not correspond to the recorded signal in the plotted example.

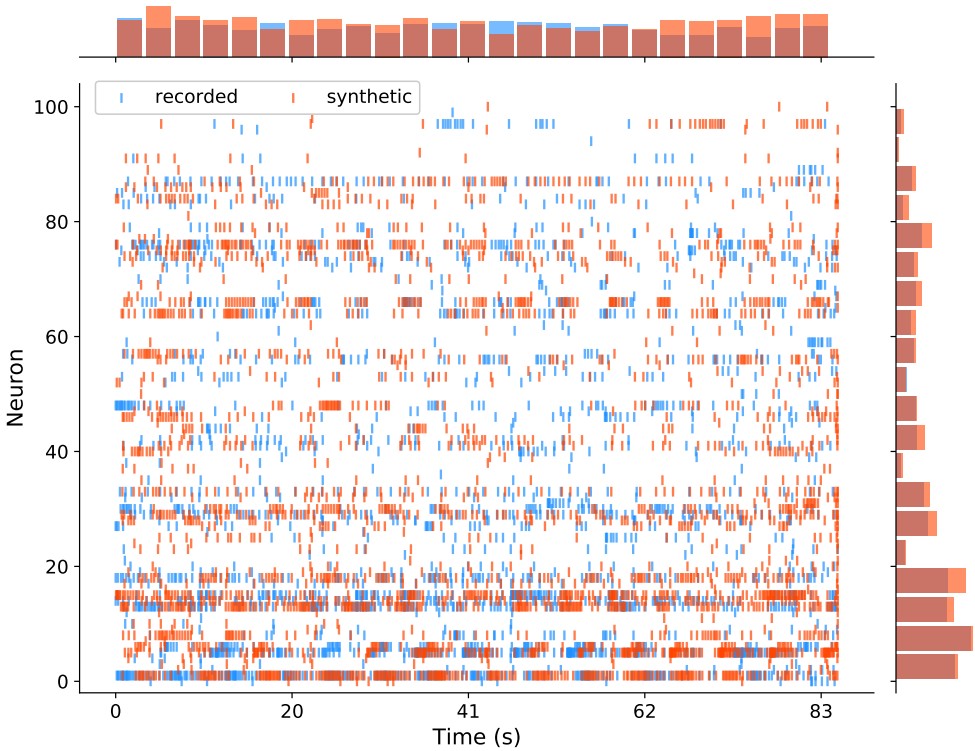

Figure A.7: Raster plot of inferred real and synthetic spike trains of a randomly selected sample generated by CalciumGAN trained on recorded data from the first day of the mice experiment.

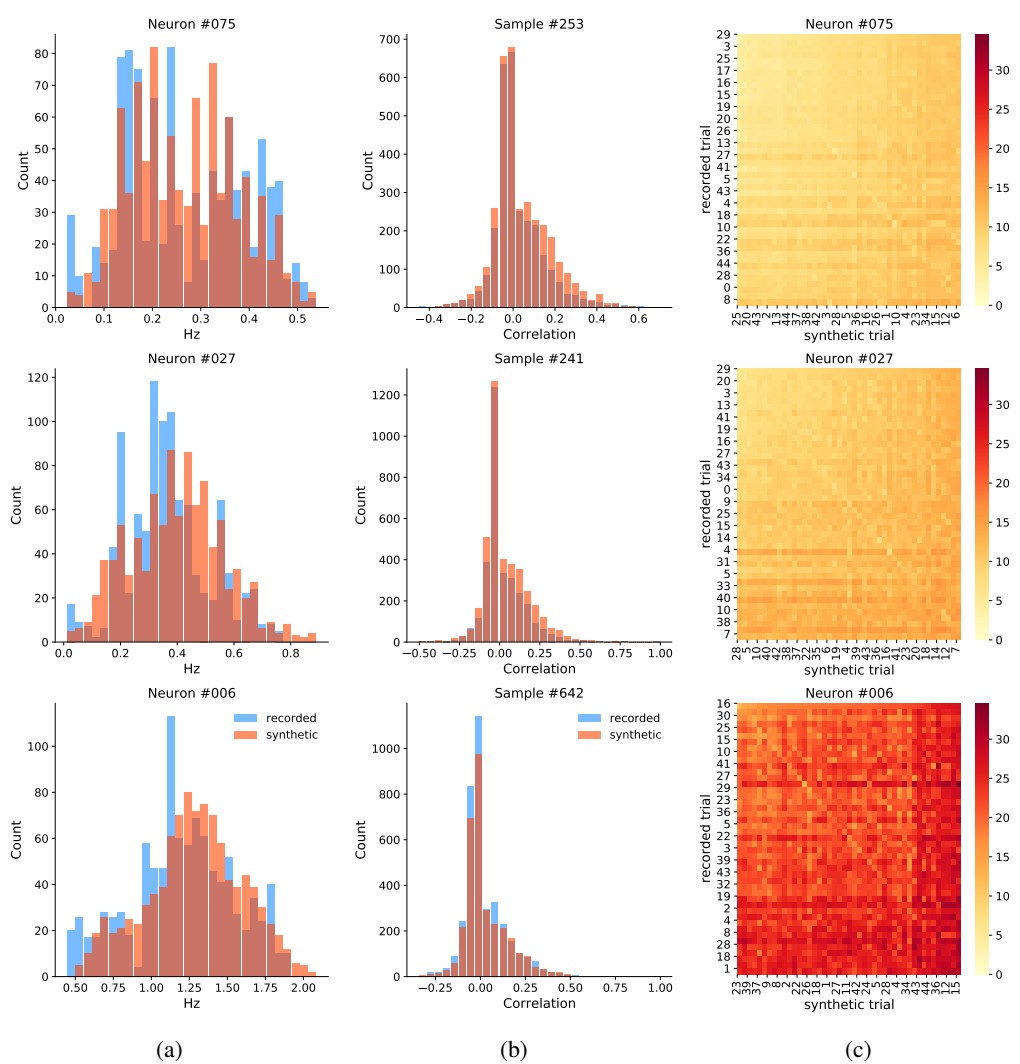

(a)          (b)          (c)

Figure A.8: First and second order statistics of data generated from CalciumGAN trained on the recorded data. Shown neurons and samples were randomly selected. (a) Mean firing rate distribution over 1000 samples per neuron. (b) Pearson correlation coefficient distribution. (c) van-Rossum distance between recorded and generated spike trains over 45 samples.

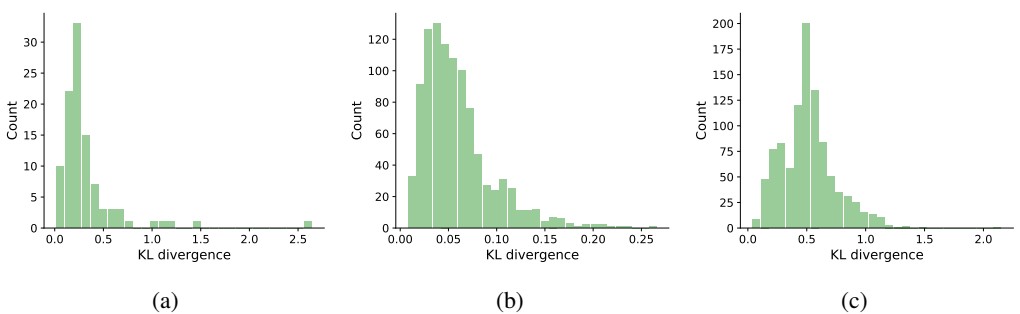

(a)          (b)          (c)

Figure A.9: KL divergence of recorded data and generated data distributions. (a) mean firing rate of each neuron, (b) pairwise Pearson correlation coefficient and (c) pairwise van-Rossum distance. The mean KL divergence of each statistics are 0.3240, 0.0590 and 0.5106 respectively.

