# OpenReview forum: "Synthesising Realistic Calcium Traces of Neuronal Populations Using GAN"
_ICLR.cc/2021/Conference — Reject_

### Official Review · AnonReviewer2 · 2020-10-27
**Effective calcium imaging trace generation but lacks demonstrated utility**

**Rating:** 3
**Confidence:** 5

**Review:**

This manuscript proposes a GAN-based approach to generate calcium traces of neurons by using adapting methods from Wasserstein GANs and WaveGAN to produce the traces.  Using OASIS to extract spikes gives similar patterns to real data on a variety of metrics. This is confirmed over a relatively large number of neurons.

The big pitfall of this manuscript is that the authors have not demonstrated the scientific utility of their method.  The purported rationale, "Generative models of neuronal activity hold the promise of alleviating the above problem by enabling the synthesis of an unlimited number of realistic samples for assessing advanced analysis methods," is not shown.  The motivation about how generative approaches can provide additional data to find higher-order relationships is fraught: perhaps the advanced analysis techniques would find more patterns by augmenting fake data, but it doesn't really increase the scientific confidence nor address the fundamental uncertainty due to limited data.  While there are some cases where research groups have used GANs to help advanced analysis techniques, such as in domain adaptation and semi-supervised learning, those cases are not shown here, nor is it clear how to build them in.  This paper needs to be revised with a clear use case to motivate it.

Pros:

The manuscript is clearly written, and shows an effective use of GANs to mimic scientific data.

Significant exploration of the results are shown.

Cons:

The metrics shown compare derived properties of the traces, not the traces themselves.  This is limiting.  In Figure 2, the real data and synthetic data have clearly different noise levels and background patterns.  If the results are visually different in a relatively small figure, it points that more exploration of the traces themselves and the generative quality needs to be explored.

The title suggests that "calcium imaging data" is being generated, but that's not really true.  Instead, the calcium traces on individual neurons are being generated, which is a derived product from calcium imaging data.  Perhaps this is just semantics, but it's misleading and should be clarified up front.

As in the major point, no clearly elucidate use case.

The lack of conditioning (on trials, behavior, unlying spike train) is a major limitation, since for most scientific applications we care about recovering these properties.

---

> ### Author Response · Authors · 2020-11-24
> **Response to Reviewer 2**
>
> We thank Reviewer 2 for their insightful comment and suggestions. We have updated the manuscript to address the concerns raised and we address the reviewer’s comments below.
> - As calcium imaging is largely being used as a proxy to measure spike trains, we have decided to evaluate and present the deconvolved spike trains. We measured the firing rate, correlation and van-Rossum distance mainly because they were used in other spike train generation models [1, 2, 3]. We would also like to point out that the model does not take trial information into consideration, hence a generated trace does not necessarily correspond to a particular recorded trace. We have updated the manuscript to clarify the choice of evaluation metrics.
> - We have updated the title and the manuscript to clarify that we are synthesizing calcium traces instead of calcium imaging data directly.
> - We were limited by the amount of recorded data collected from our live animal experiments, we weren’t able to segment our dataset based on trial information while having enough data to train the GAN model to a satisfactory level of performance. Nevertheless, preliminary results incorporating information such as trial of the experiment, location of the mice in virtual space, etc. are very promising.
>
> We would like to thank the reviewer for their time and constructive comments, we hope that the updated manuscript addresses the concerns raised by the reviewer.
>
> [1] Molano-Mazon, M., Onken, A., Piasini, E., & Panzeri, S. (2018). Synthesizing realistic neural population activity patterns using generative adversarial networks. arXiv preprint arXiv:1803.00338.
>
> [2] Ramesh, P., Atayi, M., & Macke, J. H. (2019). Adversarial training of neural encoding models on population spike trains.
>
> [3] Macke, J. H., Berens, P., Ecker, A. S., Tolias, A. S., & Bethge, M. (2009). Generating spike trains with specified correlation coefficients. Neural computation, 21(2), 397-423.

---

### Official Review · AnonReviewer3 · 2020-10-28
**Official Blind Review #3**

**Rating:** 5
**Confidence:** 4

**Review:**

Summary:
The paper proposes to use a GAN framework to generate the realistic neuronal calcium signals, enabling to scale-up the neuronal population activity data. The solution is based on WAVEGAN architecture with Wasserstein distance to train on calcium fluorescent signals. The experiments are performed in comparison to artificial calcium signals with known ground-truth closely resembles the underlying data distribution. The accuracy of the approach, robustness of generated signals from the model are evaluated.

Strengths:
- The application is very relevant as an increase in calcium imaging samples helps analyze large populations of neurons.

Weaknesses:
- This paper lacks contribution, and the novelty of the work is limited in terms of methods.
- Fig.2 does not actually show significant activation locality in the corresponding neurons; the peaks are not present in the generated signal for Neuron 27, similarly for Neuron 75 (completely mismatched with recorded data).
- This is further confirmed by Fig.3, where the generated markers are scattered across all the neurons in time, the neuronal activity irrelevant to recorded activity.
- Figures 3, 4, and 5 are not very informative as no legend is included.
- Here, the author aims to increase more realistic calcium signals for analysis of neuronal population, but how it will be useful for the healthcare/research community in terms of applications is not clear.
- In Fig.2, the generated recording does not have an inspired spike for the Neuron 27. Please clarify this.
Details for reproducing the experiments of dichotomized gaussian data are left unclear.

Major Comments:
- Why are the Pearson correlation values low (i.e., between -0.25 and 0.25)?.
- Please clarify whether the Pearson correlation is calculated between the recorded signal and the generated signal?.
- What is the performance when the model trained on real calcium signals recorded from the primary visual cortex of behaving mice and tested on artificial data?
- Suppose, if authors perturb the real calcium signals data, is it affecting the model performance?.
- Typos:
Abstract: Here we propose (Here, we propose)
Introduction: paragraph-1, high quality (high-quality)

Addressed Concerns:
- corrected the typos
- Some figures are updated

Not Addressed:
- Figure 1b is not uniform as 1a (legend information is missing)
- As mentioned in weaknesses: the novelty aspect of this work is missing.
- Further, the authors have not answered question 1 in major comments.
- Based on Fig 2 & 3, Calcium GAN generated signals are mismatching with recorded data. How one can deploy this model in real-time when model results are not identical at all?.

---

> ### Author Response · Authors · 2020-11-24
> **Response to Reviewer 3**
>
> We thank Reviewer 3 for their suggested improvements and comments. We have updated our manuscript based on the suggestions made.
>
> We address the comments below:
> - We didn’t incorporate the trial information into the model, hence the generated samples do not directly replicate a specific recorded sample. Conditioning the generator based on the trial information can be an interesting continuation of this work [1]. We have updated the manuscript and the figure descriptions to clarify this point.
> - The blue colour represents the recorded data and orange colour represents the generated data in both Figure 3 and 4. We have updated the figures and notes to clarify the plots.
> - Synthesising calcium imaging signals instead of binary spike trains eliminates the necessity for the generator to learn the deconvolution process, and calcium signals should contain more information of the neuronal activities than binary spike trains, further improving the model’s ability to learn the underlying statistics. Our method can also enable practitioners to evaluate deconvolution algorithms as the model enables them to generate synthetic calcium imaging signals with known ground-truth. Moreover, this method can potentially allow practitioners and researchers to minimize the number and duration of their live animal experiments, and instead, artificially generate (or scale-up) more calcium imaging recordings.
> - We compute the neuron pairwise Pearson correlation from each sample. Figure 4b illustrates the correlation between the recorded and generated sample from two randomly selected samples. And Figure 5b shows the KL divergence between the recorded and generated pairwise correlation of 1000 samples.
> - We are not entirely sure about the question on “What is the performance when the model trained on real calcium signals recorded from the primary visual cortex of behaving mice and tested on artificial data”, we hope that the reviewer can clarify this point.
> - We experimented on data recorded from the 1st and 4th day of the animal experiments, where the two recordings consist of very different statistics, and our model was able to learn from data with high variation.
>
> We have also addressed the typos pointed out by the reviewer.
>
> We thank the reviewer for their time and suggestions, and we hope the updated manuscript clarifies the concerns and suggestions raised by the reviewer.
>
> [1] Ramesh, P., Atayi, M., & Macke, J. H. (2019). Adversarial training of neural encoding models on population spike trains.
>
> [2] Friedrich, J., Zhou, P., & Paninski, L. (2017). Fast online deconvolution of calcium imaging data. PLoS computational biology, 13(3), e1005423.

---

### Official Review · AnonReviewer4 · 2020-10-28
**Interesting, but straightforward applications of GANs**

**Rating:** 4
**Confidence:** 4

**Review:**

In this paper, the authors construct a GAN network for generating artificial calcium imaging activity from large neural populations. This is an interesting but relatively straightforward application of GANs in a neuroscience context. Unfortunately, I don't see the pressing need for such methods, and the analysis in the paper leaves me unconvinced that the author's method works really convincingly.

* Need: The authors motivate their method with experimental hurdles to recording large ensembles of neurons, but recent years have seen a dramatic improvement in this respect (see e.g. work of Steinmetz, Pachitariu, Stringer and colleagues) who routinely record from 1000s of neurons. The authors should carefully think about what their method allows one to do that is not possible with real data.

* Evaluation: I don't understand why the authors chose to evaluate their method on the deconvolved spikes. Surely this is interesting as well, but I would have liked to see actually more of the generated data and more evaluation on the generated data directly. From Fig. 2 it seems e.g. that the noise characteristics of the generated data are quite different than for the true data. Is this the case? The DG generated data is very hard to judge, since neither the artificial or generated data are shown, just summary statistics in Fig. 1.

* Clarity: I would have liked to see a clearer explanation of what goes into the network and what comes out. Somewhere, they mention a sliding window approach - a figure would go a long way.

* What was the role of the Wasserstein objective function? Where others tried? What was their outcome?

* Table 1: How was the DG model for the experimental data estimated? Combining a normal DG with the forward model of Friedrich?


typo: p8: ref to Donahue is having () in wrong place

---

> ### Author Response · Authors · 2020-11-24
> **Response to Reviewer 4**
>
> We would like to thank Reviewer 4 for their helpful comments and suggestions. We have updated the manuscript to address some of the issues and concerns raised in the review.
>
> We address each point below:
> - The ability to generate large amounts of realistic neuronal activities of live animals, both in duration and size of the neuron population, was certainly one of the motivations for this work. In addition, synthesising calcium imaging signals instead of binary spike trains eliminate the necessity for the generator to learn the deconvolution process, and calcium signals should contain more information of the neuronal activities than binary spike trains, further improving the model’s ability to learn the underlying statistics. Moreover, our method would enable practitioners to evaluate deconvolution algorithms as the model enables them to generate synthetic calcium imaging signals with known ground-truth.
> - As calcium imaging is largely being used as a proxy to measure spike trains, we have decided to evaluate and present the deconvolved spike trains. We measured the firing rate, correlation and van-Rossum distance mainly because they were used in other spike train generation models [3, 4, 5]. We have updated the manuscript to clarify the choice of evaluation metrics.
> - The generator takes in noise sampled from standard normal and output samples that are supposed to resemble the real (recorded or synthetically generated using the DG model) preprocessed calcium traces. The discriminator takes in the generated traces or the recorded calcium traces and outputs a score on whether or not the input is real or generated data. Figure A.2 illustrates the complete pipeline of CalciumGAN. In addition, we have added Figure A.3 to illustrate the sliding window mechanism to segment the raw calcium imaging signals in our preprocessing step.
> - We used the objective function introduced in WGAN-GP to train our generator and discriminator. We have also experimented with the original GAN training objective [1], as well as LSGAN [2]. From our experiments, the WGAN-GP formulation had the best training performance and stability. We experienced model collapse and training instability (either the generator or discriminator overpower one another) with GAN and LSGAN. We have updated the manuscript to include discussions on the choice of the objective function.
> - To generate the dichotomized Gaussian (DG) data, we first fitted the DG model with the mean and covariance of the inferred spike trains from the recorded calcium imaging signals. We sample the correlated spike trains from the DG model so that it matches the duration as the recorded data (899 seconds of recordings). We then convolve the artificial spike trains to simulate calcium imaging signals (see Equation 2 and 3) as described in [2]. The artificial calcium traces were then preprocessed and trained in the same manner as the recorded data.
>
> We have also addressed the typo pointed out by the reviewer.
>
> Again, we sincerely appreciate the constructive comments made by the reviewer and please feel free to make any further suggestions so that we can improve our work.
>
> [1] Goodfellow, I., Pouget-Abadie, J., Mirza, M., Xu, B., Warde-Farley, D., Ozair, S., ... & Bengio, Y. (2014). Generative adversarial nets. In Advances in neural information processing systems (pp. 2672-2680).
>
> [2] Mao, X., Li, Q., Xie, H., Lau, R. Y., Wang, Z., & Paul Smolley, S. (2017). Least squares generative adversarial networks. In Proceedings of the IEEE international conference on computer vision (pp. 2794-2802).
>
> [3] Molano-Mazon, M., Onken, A., Piasini, E., & Panzeri, S. (2018). Synthesizing realistic neural population activity patterns using generative adversarial networks. arXiv preprint arXiv:1803.00338.
>
> [4] Ramesh, P., Atayi, M., & Macke, J. H. (2019). Adversarial training of neural encoding models on population spike trains.
>
> [5] Macke, J. H., Berens, P., Ecker, A. S., Tolias, A. S., & Bethge, M. (2009). Generating spike trains with specified correlation coefficients. Neural computation, 21(2), 397-423.

---

### Decision · Program_Chairs · 2021-01-07
**Final Decision**

**Decision:**

Reject

**Comment:**

The approach proposed here have raised major concerns from multiple reviewers especially concerning the novelty and the experimental validation procedure. Authors did not succeed in convincing reviewers of the value of their work for ML or calcium imaging processing.